# Breaking Bad:
# A Dataset for Geometric Fracture and Reassembly

**Silvia Sellán**[1,*] **Yun-Chun Chen**[1,2,*] **Ziyi Wu**[1,2,*] **Animesh Garg**[1,2,3] **Alec Jacobson**[1,2,4]

[1]University of Toronto [2]Vector Institute [3]NVIDIA [4]Adobe Research, Toronto

`{sgsellan, ycchen, ziyiwu, garg, jacobson}@cs.toronto.edu`

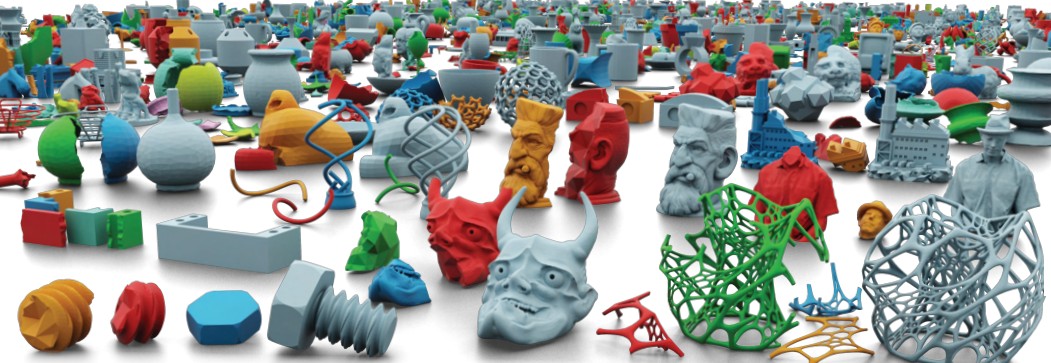

Figure 1: **The Breaking Bad dataset** contains over one million fractured objects. A subset of these is shown here, each base model in light blue and each fractured piece in a different color. This dataset can be used for machine learning applications such as geometric reassembly or example-based fracture simulation.

## Abstract

We introduce Breaking Bad, a large-scale dataset of fractured objects. Our dataset consists of over one million fractured objects simulated from ten thousand base models. The fracture simulation is powered by a recent physically based algorithm that efficiently generates a variety of fracture modes of an object. Existing shape assembly datasets decompose objects according to semantically meaningful parts, effectively modeling the *construction* process. In contrast, Breaking Bad models the *destruction* process of how a geometric object naturally breaks into fragments. Our dataset serves as a benchmark that enables the study of fractured object reassembly and presents new challenges for geometric shape understanding. We analyze our dataset with several geometry measurements and benchmark three state-of-the-art shape assembly deep learning methods under various settings. Extensive experimental results demonstrate the difficulty of our dataset, calling on future research in model designs specifically for the geometric shape assembly task. We host our dataset at https://breaking-bad-dataset.github.io/.

## 1 Introduction

Fracture reassembly aims to compose the fragments of a fractured object back into its original shape, e.g., a shattered sculpture or a broken item of kitchenware. With applications in artifact preservation [38, 40], digital heritage archiving [39, 43], computer vision [20, 29], robotics [12, 24] and geometry processing [2, 21], fracture reassembly is a practical yet challenging task that receives attention from multiple communties.

---

*Equal contribution

36th Conference on Neural Information Processing Systems (NeurIPS 2022) Track on Datasets and Benchmarks.

Table 1: Our Breaking Bad dataset contains a large number of fractured objects with various, physically realistic fractures. S: shapes. BP: breakdown patterns. OP: object parts.

| Dataset | #S | #BP | #BP / #S | #OP / #BP | Decomposition | Physically based |
|---|---|---|---|---|---|---|
| PartNet [32] | 26,671 | 26,671 | 1 | 21.51 | Semantic | No |
| AutoMate [23] | 92,529 | 92,529 | 1 | 5.85 | Semantic | No |
| JoinABLe [59] | 8,251 | 8,251 | 1 | 18.72 | Semantic | No |
| NSM dataset [7] | 1,246 | 201,590 | 161.78 | 2 | Geometric | No |
| **Breaking Bad** | 10,474 | **1,047,400** | 100 | 8.06 | Geometric | Yes |

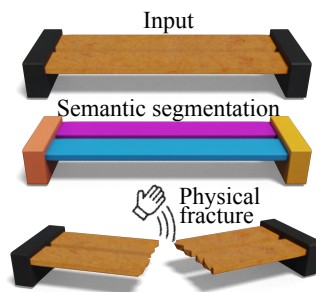

Machine learning approaches have shown progress on the fracture reassembly task [10, 55], but require large-scale datasets of fracture objects. Existing assembly datasets like PartNet [32], AutoMate [23] and JoinABLe [59] are constructed based on human or automated semantic segmentation. The objects in these datasets are decomposed in a semantically consistent way. However, objects that break naturally due to external forces generally do not break into fragments that are semantically well defined (see inset). Thus, existing part assembly datasets are not suitable for studying fracture reassembly.

Simulating how an object fractures when receiving an impact is a well-studied scientific problem. One can generate a dataset of broken objects by using any existing algorithm (e.g., [36, 61]) to simulate how objects in a typical shape dataset would break under randomly sampled dynamic conditions. Unfortunately, the high computational cost of physics-based fracture simulation algorithms (e.g., those used in engineering or the film industry) makes them hard to scale. While fast fracture algorithms (e.g., those for real-time applications like video games) exist to account for this shortcoming [33, 54], they use consistent geometric strategies to produce fracture patterns with no physical realism guarantee, limiting dataset diversity and generalization.

Recently, Sellán et al. [51] introduced the concept of an object's *fracture modes*, which correspond to a shape's most geometrically natural form of breaking apart. Once these modes are precomputed for a given object, different impacts can be projected onto the modes to produce different fracture patterns in milliseconds. By producing physically realistic, diverse breaking patterns with a reasonably fast runtime, this method provides a good tradeoff for fractured object data generation.

In this paper, we introduce Breaking Bad, a large-scale fractured object dataset. We collect base models from Thingi10K [65] and PartNet [32] and apply Sellán et al.'s fracture simulation algorithm to each. For each base model, we compute the first 20 fracture modes, generating 20 fracture patterns. Using these modes, we sample 80 additional random impacts and project them onto these fracture modes to generate 80 additional fracture patterns. This results in a total of 100 unique fracture patterns per base model. Our dataset contains a diverse set of shapes spanning everyday objects, artifacts, and objects that are commonly used in video gaming, fabrication, and example-based fracture simulation, combining one million geometrically natural fracture patterns (see Figure 1).

Breaking Bad is a suitable dataset for studying the reassembly task and presents several challenges to candidate solutions, including complex shape geometry, large variations in fracture volumes, and varying numbers of fractured pieces per shape. We analyze Breaking Bad with several geometry measurements and benchmark three state-of-the-art deep learning models under various settings. Extensive experiments against Breaking Bad reveal that fractured shape reassembly is still a quite open problem, inviting opportunities for future contributions.

**Summary of contributions:**

1. We introduce a large-scale dataset of fractured objects for the geometric shape assembly task.

2. We provide a geometric analysis of the collected dataset.

3. We benchmark three state-of-the-art deep learning methods on our dataset under various settings, with accompanying code to ensure reproducibility and facilitate future research.

4. Our dataset is publicly available at `https://breaking-bad-dataset.github.io/` (see prototype website in Figure 4).

Table 2: Dataset statistics at different percentiles for each subset in our dataset. O: objects. FP: fractured pieces. V: vertices. F: faces. PCR: piece convexity rank.

| Category | #O | #FP / #O | | | #V / #FP | | | #F / #FP | | | V / #FP ($\times 10^{-4}$) | | | PCR ($\times 10^{-2}$) | | |
|---|---|---|---|---|---|---|---|---|---|---|---|---|---|---|---|---|
| Percentile | | 25th | 50th | 75th | 25th | 50th | 75th | 25th | 50th | 75th | 25th | 50th | 75th | 25th | 50th | 75th |
| Everyday | 542 | 2 | 3 | 6 | 98 | 279 | 949 | 216 | 742 | 3,162 | 2.47 | 9.32 | 71.13 | 6.19 | 16.63 | 44.00 |
| Artifacts | 204 | 3 | 8 | 19 | 90 | 307 | 757 | 208 | 872 | 2,310 | 0.64 | 3.96 | 23.75 | 9.13 | 19.78 | 45.44 |
| Others | 9,475 | 3 | 6 | 13 | 70 | 331 | 1,119 | 146 | 832 | 3,222 | 0.51 | 6.17 | 39.36 | 5.58 | 11.17 | 16.00 |
| All | 10,221 | 3 | 10 | 13 | 92 | 286 | 1,345 | 22 | 286 | 2,552 | 0.05 | 2.70 | 31.89 | 6.38 | 13.99 | 28.90 |

## 2 Related Work

We review prior work and its relationship to our choices, focusing on works relevant to our target applications, dataset scope, or data generation.

**3D shape assembly.** Shape assembly has been widely studied (see inset on the right for a typical vision-based shape assembly pipeline). Existing methods studying part assembly [10, 16, 20, 23, 29, 55, 59, 63] aim at composing a complete object from a set of parts by leveraging part segmentation [29] or formulate part assembly as a graph learning problem [16, 20]. These methods use the

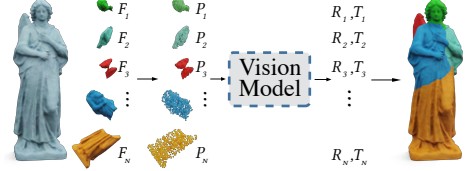

Object Pieces Points      Poses Assembly

PartNet dataset [32] in which the objects are decomposed in a semantically consistent way. Unlike these works, we consider the task of reassembling fractured objects, where fracture patterns are physically realistic yet semantically inconsistent. NSM [7] is the one that comes closest to our task setting. However, NSM is designed for two-part assembly, whereas objects in our dataset often break into multiple fractured pieces (on average 8.06 pieces per fracture).

**Shape assembly datasets.** Early shape assembly datasets [5, 10, 21, 52] are often small in size and contain only a few categories. To develop learning-based algorithms for shape assembly, several large-scale datasets have been constructed [7, 23, 32, 59]. PartNet [32], AutoMate [23] and JoinABLe [59] are datasets that contain shapes where the object decomposition is semantically consistent. The object breakdown patterns in these datasets are not suited for the fractured shape reassembly task, where objects generally do not break in semantically meaningful ways. The object decomposition patterns in the NSM dataset [7] are not determined by a semantic part decomposition. Instead, they trim models with a non-predefined set of parametric functions (e.g., a sine function). This arbitrary decomposition will in general bear no relationship to the physically meaningful fracture behavior of a given object. In contrast, our dataset is generated by a physically based fracture simulation algorithm [51], which generates various fracture patterns of a single object. See Table 1 for a comparison between datasets.

**Fracture simulation.** Computing the fracture pattern of a base shape under certain conditions is a well-studied problem for its applications in physics, engineering, and computer graphics. In graphics, previous work has focused on producing realistic-looking fractures in runtimes suitable for their use in the film and video game industries. These can be grouped into physical and procedural methods.

Physical fracture simulation methods model the dynamic growth of fracture faults at very high temporal and spatial resolutions, with discretization strategies that vary from mass-springs [18, 35] to finite elements [25, 27, 36, 42, 58], boundary elements [14, 15, 66], and the material-point method [60, 61]. Unfortunately, the realistic-looking results produced by these algorithms require significant runtimes, often in the days or weeks for a single simulation. While these costs can be assumed by a film studio seeking to produce the perfect breaking scene, they make these methods ill-suited for dataset generation at a massive scale.

Procedural fracture algorithms use geometric heuristics to precompute a *prefracture* of an object into realistic-looking pieces before simulation. This can be achieved, for example, by cutting a shape by the Voronoi diagrams of randomly scattered points [37, 45], perturbed level-set functions (e.g., [7, 34]) or pre-authored fracture patterns [33, 54]. They then use geometric strategies such as Euclidean distance thresholds to decide which fractures get activated when a sufficiently strong contact is detected. While some of these algorithms are fast enough to be used at a massive scale, the artificial regularity of the produced pieces (e.g., Voronoi diagrams produce only convex patterns) would limit the diversity of any produced dataset. Further, the fact that the fracture patterns are not physically based would make it harder for learning from data to generalize to real-world scenarios.

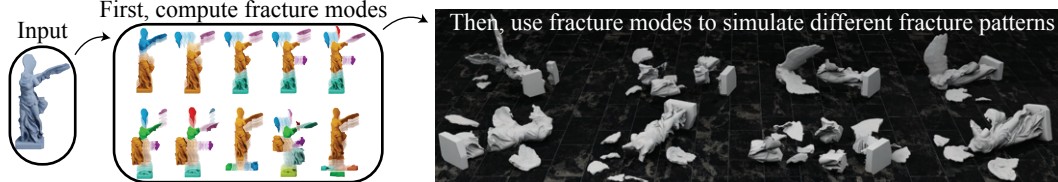

Figure 2: **Dataset processing pipeline.** For each shape in our base dataset (*left*), we first compute its fracture modes (*middle*) and then use them to simulate many different fracture patterns (*right*).

Recently, Sellán et al. [51] bridged the gap between physical and procedural fracture by providing a physics-based pre-fracture algorithm. We adopt this method for our dataset generation task, as it efficiently (i.e., in a scalable way) computes a set of orthogonal (i.e., maximally different), natural (i.e., likely) fracture patterns. We review their algorithm below.

## 3   Background: Fracture Modes for Fast Simulation

Let $\Omega$ be a volumetric mesh with $n$ vertices and $m$ tetrahedra. The first $k$ *elastic modes* of $\Omega$ are then defined as the columns of a matrix $\mathbf{U} \in \mathbb{R}^{3n \times k}$ which solve the eigenvalue problem:

$$\underset{\mathbf{U}^\top \mathbf{M} \mathbf{U} = \mathbf{I}}{\operatorname{argmin}} \quad \frac{1}{2} \sum_{r=1}^{k} \operatorname{trace} \left( \mathbf{U}^\top \mathbf{Q} \mathbf{U} \right), \tag{1}$$

where $\mathbf{Q}$ is the Hessian matrix of some elastic strain energy at the rest configuration and $\mathbf{M}$ is the traditional FEM mass matrix.

Sellán et al. [51] generalize this idea to generate *discontinuous* "fracture modes." The key insight is to define solutions over a larger discontinuous function space, where values are not stored per vertex, but rather per corner of each tetrahedron. Their fracture modes are columns in a matrix $\tilde{\mathbf{U}} \in \mathbb{R}^{12m \times k}$ which solve the *generalized* eigenvalue problem:

$$\underset{\tilde{\mathbf{U}}^\top \tilde{\mathbf{M}} \tilde{\mathbf{U}} = \mathbf{I}}{\operatorname{argmin}} \quad \frac{1}{2} \sum_{r=1}^{k} \operatorname{trace} \left( \tilde{\mathbf{U}}^\top \tilde{\mathbf{Q}} \tilde{\mathbf{U}} \right) + \omega \sum_{r=1}^{k} E_D(\tilde{\mathbf{u}}_r), \tag{2}$$

where the $\tilde{\mathbf{Q}}, \tilde{\mathbf{M}} \in \mathbb{R}^{12m \times 12m}$ operators are trivial extensions into this larger function space, $E_D(\tilde{\mathbf{u}}_r)$ is a convex objective function measuring the total amount of discontinuity (intuitively, how fractured the function is), and $\omega$ is a weight balancing both terms. As shown by Sellán et al. [51], the value of $\omega$ has no effect on the output modes as long as it is small enough, so we fix it at $0.001$.

The computed fracture modes can then be used for efficient destruction simulation. Any impact on the shape represented as a vector $\mathbf{w} \in \mathbb{R}^{12m}$ can be *projected* onto the precomputed fracture modes to obtain a fractured displacement of the model:

$$\mathbf{w}^\star = \tilde{\mathbf{U}} \tilde{\mathbf{U}}^\top \tilde{\mathbf{M}} \mathbf{w}. \tag{3}$$

Like the individual modes, $\mathbf{w}^\star$ will represent the function with discontinuities. By identifying these (through a discontinuity threshold $\tau$), one can compute an impact-dependent fracture pattern for $\Omega$. Much of the computational cost of this impact projection step can benefit from precomputation, as shown in [51]. Thus, the impact-specific runtime has linear complexity in the mesh size. This precomputation also benefits our dataset generation by allowing very efficient storage of many fractures of the same shape (see Section 4.3). We refer the reader to [51] for more details about the fracture simulation algorithm.

## 4   The Breaking Bad Dataset

Our dataset contains results of fracture simulation conducted on a large base library of 3D shapes.

**Base shape selection.** Since our main intended application is to facilitate research in shape reassembly, we first collect all meshes from 20 daily object categories in PartNet [32], i.e., BeerBottle, Bottle,

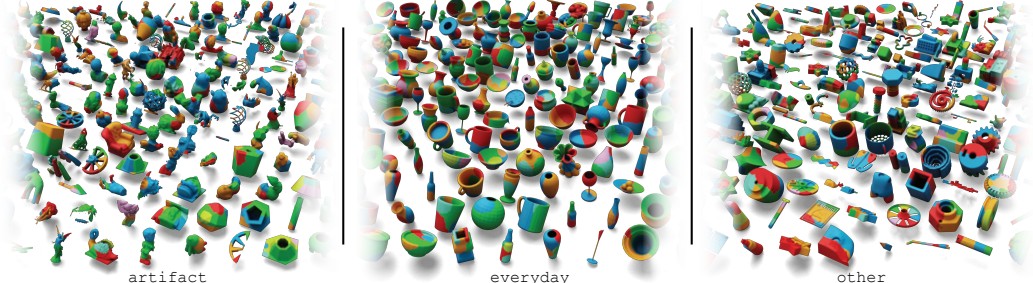

Figure 3: **Dataset gallery.** Our dataset contains a diverse set of objects, which can be used for various applications. (*Left*) Artifacts for archaeology applications. (*Middle*) Everyday objects for computer vision and robotics applications. (*Right*) Other objects for video gaming and example-based fracture simulation.

Bowl, Cup, Cookie, DrinkBottle, DrinkingUtensil, Mirror, Mug, PillBottle, Plate, Ring, Spoon, Statue, Teacup, Teapot, ToyFigure, Vase, WineBottle, and WineGlass, forming the `everyday` object subset. We also select meshes with tag sculpture and tag scan from Thingi10K [65] to construct the `artifact` subset, which contains common objects in archeology. Finally, we construct the `other` subset using all the remaining meshes from Thingi10K to increase the diversity of our dataset. For Thingi10K models, we use the pre-processed watertight meshes provided by Hu et al. [19].

## 4.1 Fracture Simulation

Figure 2 presents the dataset processing pipeline. We treat objects as solids and assume isotropic materials. We process each of the selected input meshes in the base dataset independently. We re-scale each of them to fit a unit-length box for parameter choice consistency. This normalization scheme allows our method to be *scale invariant*.

We begin by constructing a coarse (4,000 face) cage triangular mesh that fully contains the input following the Simple Nested Cages (SNC) algorithm introduced in [51], with a grid size of 100. While more tightly-fitting cage generation algorithms exist (e.g., [47]), we find SNC, which relies on signed distance computation [4] and isosurface extraction [30], to be superior in robustness, runtime and reliability. These are all aspects that are critical for geometry processing at a large scale. We tetrahedralize the cage using TETGEN [53].

We compute the first 20 fracture modes of the tetrahedral cage mesh following the scheme described in Section 3. We then transfer these modes to the input mesh, intersecting all the possible fracture patterns that can be spanned by the modes with the input mesh as described in Section 3.5 of [51]. This step is the performance bottleneck of our processing pipeline, covering around 70% of runtime.

After this pre-computation step, we simulate impacts at random points on the cage geometry's surface, obtaining each impact-dependent fracture pattern in around one millisecond. For each impact, we also randomize the discontinuity threshold $\tau$ to account for different materials being more or less prone to breaking. We discard a fracture pattern if it produces fewer than two or more than 100 pieces (while perhaps realistic behavior, this can make the reassembly task excessively difficult) and repeat until we have 80 valid fracture patterns. We add these, together with the 20 fracture modes, to our dataset, totaling 100 fracture patterns per base shape. The speed of the impact projection makes it so that this step is dominated by the writing of the output fractured meshes into our dataset. See inset for examples of fractured objects in each subset of our dataset.

**Implementation.** We implement our data processing pipeline in PYTHON, using LIBIGL [22] for common geometry processing subroutines. We run the jobs on a dedicated CPU cluster with 2.5GHz Intel(R) Xeon(R) processors. We use 320 CPU cores, each with 386GB RAM, to parallelize the data generation jobs. With our efficient simulator, we can generate the entire dataset in 24 hours.

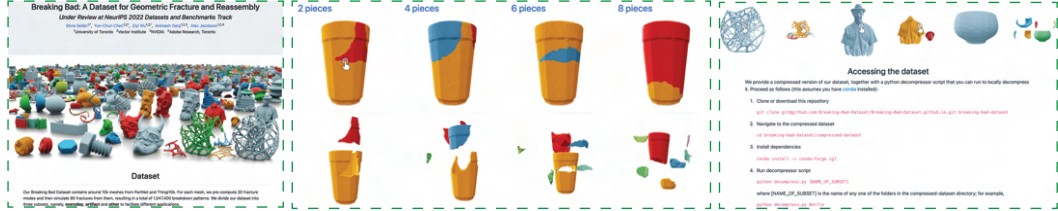

Figure 4: **Dataset acccess.** We host our dataset at `https://breaking-bad-dataset.github.io/`, which includes a gallery for exploring individual fractured objects and shape assembly results interactively and instructions for accessing the dataset.

## 4.2 Dataset Analysis

We report statistics at the 25th, 50th, and 75th percentiles of a variety of geometric characteristics: report the number of fractured pieces per object, the number of vertices per fractured piece, the number of faces per fractured piece, the volume per fractured piece for each subset in Table 2. A tell-tale sign of prior precomputed fracture methods (e.g., [33]) is the overabundance of convex pieces. We quantify the convexity of our produced fracture pieces using the rank defined by Asafi et al. [3] (computed on a randomly selected subset of 1,000 pieces). The inset presents the percentile plot of fractured pieces of each subset. Our dataset has a wide distribution over the number of fractured pieces with large variations of volume. See Figure 3 for some examples of fractured objects in each subset of our dataset.

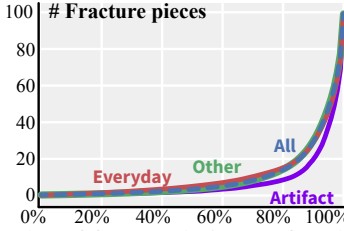

## 4.3 Dataset Access and Storage

Our full dataset contains over one million individual fractured shapes. Stored in a standard geometric file format (i.e., .OBJ), it occupies well over 1TB (before zipping). While its massive scale is one of the main contributions of our dataset, it can also complicate its sharing and storage. To address this, we draw from the specifics of our fracture simulation step to produce a losslessly *compressed* version of our dataset, which contains all the same information as our full dataset in as little as 10GB before zipping and 7.3GB after zipping. We release it alongside our full dataset.

By nature of the projection step in Equation (3), the simulated fractures can only contain discontinuities at fracture faults that are already present in at least one of the fracture modes. Therefore, we can use the fracture modes to compute a super-segmentation of the base shape into all possible pieces that can result from projecting impacts onto them. Thus, instead of storing 100 fracture patterns per base shape, we can store only this super-segmentation and, for each projected impact, per-piece labeling identifies which pieces break off. This reduces the size of our dataset by almost a factor of 100. Decompressing the data is just a matter of looping over all fractures and pasting the pieces of the super-segmented mesh that do not break off in each case. We provide a PYTHON script that does this.

We develop a website (see Figure 4) to host our dataset and facilitate interactive exploring of the fractured objects. Our prototype at `https://breaking-bad-dataset.github.io/` contains the compressed dataset, the decompression instructions, and user-friendly gallery views, allowing direct download of individual models.

## 4.4 Licensing

We gather our base models following the licenses specified in each of the source datasets: the MIT license in the PARTNET dataset and a variety of open-source licenses in the THINGI10K dataset (see Figure 12 in [65]). We release each model in our dataset with an as-permissive-as-possible license compatible with its underlying base model and all code under the MIT license.

## 5 Case Study Application: 3D Geometric Shape Assembly

The Breaking Bad dataset can be used for applications in the 3D geometric shape assembly task. In this section, we consider vision-based 3D geometric shape assembly as a case-study application.

Table 3: **Evaluation on fracture reassembly.** We report the results of three learning-based shape assembly models on the `everyday` object subset. The results are averaged over all 20 categories.

| Method | RMSE ($R$) ↓ | RMSE ($T$) ↓ | CD ↓ | PA ↑ |
|--------|-----------|-----------|------|------|
|        | degree | $\times 10^{-2}$ | $\times 10^{-3}$ | % |
| Global | 80.7 | 15.1 | 14.6 | 24.6 |
| LSTM   | 84.2 | 16.2 | 15.8 | 22.7 |
| DGL    | 79.4 | 15.0 | 14.3 | 31.0 |

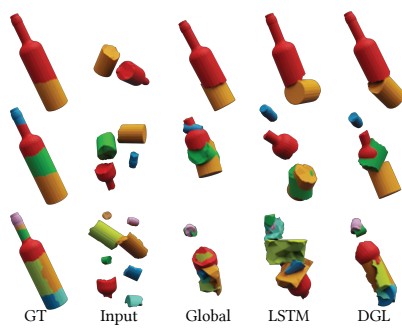

Figure 5: Visual results on the `everyday` object subset.

Following existing methods [7, 20], we assume the models only have access to point clouds sampled from each fracture. Meshes are only used for result visualizations. Given $N$ mesh fractured pieces of an object $\mathcal{F} = \{F_i\}_{i=1}^{N}$, we sample a point cloud from the mesh of each fractured piece forming $\mathcal{P} = \{P_i\}_{i=1}^{N}$, where $N$ is the number of fracture pieces which varies across different shapes and fracture patterns. We aim to learn a model that takes as input the sampled point clouds $\mathcal{P}$ and predicts the canonical SE(3) pose for each point cloud (fractured piece). Denote the predicted SE(3) pose of the $i$-th fractured piece as $q_i = \{(R_i, T_i) \mid R_i \in \mathbb{R}^{3\times3}, T_i \in \mathbb{R}^3\}$, where $R_i$ is the rotation matrix and $T_i$ is the translation vector. We apply the predicted SE(3) poses to transform the pose of each fractured piece and get $q_i(P_i) = R_i P_i + T_i$, respectively. The union of all of the pose-transformed point clouds $S = \bigcup_i q_i(P_i)$ results in the predicted assembly result.

We benchmark three state-of-the-art shape assembly methods and perform analysis on the Breaking Bad dataset to answer the following questions:

1. How do learning-based shape assembly methods perform on fracture reassembly? (Section 6.1)

2. How does the number of fractured pieces affect performance? (Section 6.2)

3. Do model pre-training and fine-tuning schemes help improve performance? (Section 6.3)

4. How well do state-of-the-art shape assembly methods generalize to unseen objects? (Section 6.4)

**Evaluation metrics.** Following the same evaluation scheme as NSM [7], we compute the root mean square error (RMSE) and the mean absolute error (MAE) between the predicted and ground-truth rotation $R$ and translation $T$, respectively. We use Euler angle to represent rotation. We report the RMSE results in the main paper and the results of MAE are provided in the Appendix. In addition, we follow the evaluation protocol in [29] and adopt the shape chamfer distance (CD) and part accuracy (PA) metrics for performance evaluation. The details of shape chamfer distance and part accuracy are provided in Appendix B.

**Baseline methods.** We select three state-of-the-art learning-based shape assembly methods for benchmarking performance on our dataset: Global [28, 48], LSTM [62] and DGL [20]. We note that while NSM [7] is a learning-based method that is designed for geometric shape assembly, their model only considers two-part assembly, which cannot be applied directly to address the multi-part assembly problem, which is the task considered in this paper. We also do not benchmark RGL-NET [16] on our dataset, because it requires part ordering information for shape assembly. While such an ordering can be defined in semantic part assembly (e.g. chair seat → chair leg → chair back), the definition is unclear in geometric shape assembly. Without the ordering information, RGL-NET will degenerate to DGL. We therefore do not include NSM and RGL-Net for benchmarking performance. The details of each baseline method are provided in Appendix C.

**Implementation details.** Meshes in each category are aligned to a canonical pose as the ground-truth assembly. In each experiment, we use 80% data for training and the remaining 20% for testing. We sample 1,000 points from each fractured piece on the fly during training. Each point cloud is zero-centered and randomly rotated, providing self-supervision labels for model training. More training and implementation details are summarized in Appendix E.

# 6 Case Study Evaluation

With our experimental setup in place, we now evaluate existing shape assembly methods using the Breaking Bad dataset and probe these methods along axes afforded by our dataset's characteristics.

## 6.1 Task Performance

Following semantic part assembly methods [20, 29], we train and test each baseline method on fractured objects with at most 20 fractured pieces, train one model for each category, and report performance averaged over all 20 categories. We use the `everyday` object subset for evaluation.

Table 3 presents the performance of three baseline methods. Quantitatively, DGL outperforms Global and LSTM on all metrics. Qualitatively, (see Figure 5) DGL predicts assembly results that are visually more similar to ground truths compared to the other two baselines. The results suggest that GNNs have a better ability in reasoning about the fit between fractures than the other two architectures.

While these methods achieve strong performance on the semantic part assembly task, they all suffer from a drastic performance drop on the geometric shape assembly task (the task considered in this paper). This is because in semantic shape assembly, all parts have clear semantic meanings and shape assembly can be achieved by leveraging such priors. To achieve this, existing methods use PointNet [44] to learn a global shape feature for each part. In contrast, geometric shape assembly has to rely on learned local features for local surface matching, which cannot be achieved by PointNet. The significant performance drop highlights the difference between semantic shape assembly and geometric shape assembly and suggests that specific model designs leveraging local geometric cues for assembling fractured pieces are required. More results are provided in Appendix F.1.

## 6.2 Ablation Study: Number of Fractured Pieces

Existing semantic part assembly methods [20, 28, 48, 62] only consider cases where each object has at most 20 parts. However, objects can break into more numbers of fractured pieces. Since our dataset contains objects with up to 100 fractured pieces, we analyze how training with different numbers of fractured pieces affects model performance. Specifically, we train DGL on the `everyday` object subset with three settings: (i) training the model on fractured objects with 2 to 20 fractured pieces, (ii) 2 to 50 fractured pieces, and (iii) 2 to 100 fractured pieces. In each setting, we report the performance evaluated on objects with 2 to 20, 21 to 50, and 51 to 100 fractured pieces, respectively.

As shown in Table 4, the performance evaluated by chamfer distance and part accuracy drops significantly when the model is tested on objects with more numbers of fractures. Training on more numbers of fractures improves chamfer distance and part accuracy evaluated on objects with 21 to 50 and 51 to 100 fractures. Shape assembly is a combinatorial problem. As the number of fractures increases, the problem complexity increases. The ablation study results concur with this and demonstrate the difficulty of our dataset. More quantitative results are provided in Appendix F.2.

## 6.3 Analysis of Model Pre-training and Fine-tuning

Model pre-training and fine-tuning have been shown effective in many vision tasks [17, 46, 67]. We analyze how applying model pre-training and fine-tuning schemes affect performance on the fracture reassembly task. We adopt the `artifact` subset for quantifying performance. Following Section 6.1, we train and test each baseline method on objects with at most 20 fractured pieces.

We report the results of training each baseline from scratch in the top block of Table 5 and those obtained by fine-tuning from the respective models in Table 3 in the bottom block of Table 5. All three models improve chamfer distance and part accuracy when the models are fine-tuned from the respective models pre-trained on the `everyday` object subset. This finding is in line with those in recent model pre-training studies [17, 46, 67]. More quantitative results are provided in Appendix F.3.

## 6.4 Generalization to Unseen Objects

A core question in machine learning and computer vision is generalization. We investigate this and ask how well do the three learning-based shape assembly methods generalize to unseen objects.

Table 4: **Ablation study: Number of fractured pieces.** We train DGL on the `everyday` object subset in three settings and report performance evaluated on fractured objects of different numbers of fractured pieces. The results are averaged over all 20 categories.

| Test set | RMSE $(R) \downarrow$ | RMSE $(T) \downarrow$ | CD $\downarrow$ | PA $\uparrow$ |
|---|---|---|---|---|
| pieces | degree | $\times 10^{-2}$ | $\times 10^{-3}$ | % |
| Results of training on fractured objects with 2 to 20 fracture pieces | | | | |
| 2-20 | 79.4 | 15.0 | 14.3 | 31.0 |
| 21-50 | 84.4 | 20.1 | 15.0 | 7.5 |
| 51-100 | 85.1 | 21.3 | 23.0 | 4.8 |
| Results of training on fractured objects with 2 to 50 fracture pieces | | | | |
| 2-20 | 79.9 | 14.8 | 14.0 | 29.9 |
| 21-50 | 84.5 | 19.6 | 14.0 | 7.7 |
| 51-100 | 84.8 | 20.5 | 18.1 | 4.7 |
| Results of training on fractured objects with 2 to 100 fracture pieces | | | | |
| 2-20 | 79.8 | 14.4 | 14.0 | 29.4 |
| 21-50 | 84.3 | 19.2 | 14.5 | 7.4 |
| 51-100 | 85.3 | 20.0 | 13.9 | 4.8 |

Table 5: **Analysis of model pre-training and fine-tuning.** We report the results of three learning-based shape assembly models on the `artifact` subset.

| Method | RMSE $(R) \downarrow$ | RMSE $(T) \downarrow$ | CD $\downarrow$ | PA $\uparrow$ |
|---|---|---|---|---|
| | degree | $\times 10^{-2}$ | $\times 10^{-3}$ | % |
| Results of training the model from scratch | | | | |
| Global | 84.8 | 16.7 | 19.0 | 12.7 |
| LSTM | 85.2 | 17.2 | 23.5 | 6.6 |
| DGL | 85.8 | 16.8 | 19.4 | 12.8 |
| Results of fine-tuning from the model in Table 3 | | | | |
| Global | 83.8 | 16.6 | 19.0 | 13.3 |
| LSTM | 84.6 | 16.8 | 21.5 | 11.7 |
| DGL | 81.7 | 16.6 | 17.3 | 19.4 |

Table 6: **Generalization to unseen objects.** We report the results of three learning-based shape assembly models on the `other` subset.

| Method | RMSE $(R) \downarrow$ | RMSE $(T) \downarrow$ | CD $\downarrow$ | PA $\uparrow$ |
|---|---|---|---|---|
| | degree | $\times 10^{-2}$ | $\times 10^{-3}$ | % |
| Results of testing the model in Table 3 | | | | |
| Global | 86.4 | 19.4 | 42.2 | 6.0 |
| LSTM | 84.9 | 18.7 | 45.3 | 4.8 |
| DGL | 86.6 | 20.1 | 38.5 | 7.5 |
| Results of testing the model in the bottom block of Table 5 | | | | |
| Global | 83.9 | 18.8 | 39.2 | 6.7 |
| LSTM | 82.9 | 17.9 | 40.3 | 5.5 |
| DGL | 81.3 | 17.2 | 36.6 | 8.3 |

We take the models in Table 3 (models trained on the `everyday` object subset) and the models in the bottom block of Table 5 (models trained on the `everyday` object subset and fine-tuned on the `artifact` subset), and test them on the `other` subset. Similar to Section 6.1, we train and test each baseline method on fractured objects with at most 20 fractured pieces.

We report the results of testing the model in Table 3 in the top block of Table 6 and the results of testing the model in the bottom block of Table 5 in the bottom block of Table 6. Both chamfer distance and part accuracy become worse when compared to the results in Table 3 and Table 5. This is not surprising as the models are never trained on the `other` subset and the shape geometry and fracture patterns in the `other` subset are different from those in the `everyday` object and `artifact` subsets (see Figure 3). More quantitative results are provided in Appendix F.4.

While the results show that all three learning-based shape assembly models do not generalize well, we observe that when comparing the results between the top and bottom blocks of Table 6 the models pre-trained on more data (models used in the bottom block) achieve better chamfer distance and part accuracy results. The finding here is consistent with those in computer vision tasks that training on more data results in better model generalization [6, 8, 13].

## 7 Limitations & Future Work

We inherit all the fundamental limitations of our choice of the underlying fracture simulation method [51]. Most notably, fractures are assumed to be brittle (as opposed to ductile). This is reasonable for stiff materials like ceramics, glass or plastic undergoing sudden impacts, but not representative of fractures caused by extensive plastic deformations (e.g., when bending or

stretching rubber or metal until finally reaching its breaking point). Further, fractures are assumed to instantaneously appear. In reality, fractures propagate through a shape according to relieved stress. One characteristic is that faults tend to be perpendicular at junctures. This quality is absent from Sellán et al.'s fracture patterns; known methods for achieving this require excessively small simulation time-stepping, prohibiting efficient construction of a large-scale dataset generation. Finally, our fractures follow the faces of the underlying tetrahedral mesh. Sellán et al. suggest post-processing fracture surfaces with the upper-envelope-based method of Abdrashitov et al. [1]. As this post-processing introduces yet another hyperparameter, we leave smoothing as an option for the dataset user to conduct on their own. Future improvements of our dataset could alternate between [51] and other fracture algorithms at the generation stage to mitigate simulation-specific biases.

While Sellán et al. [51] can simulate material changes and fracture anisotropies through a user-specified vector field, setting this parameter automatically given an object is a research problem beyond the scope of our work. For scalability, we instead assume every object to be made of a single material with no prefered fracture direction. Future work could couple our fracture generation with neural material or semantic segmentations to produce fractures for more complex objects.

Our choice of the base model library includes models relevant to many situations, but would be inappropriate for others (e.g., medical domains requiring anatomical accuracy). We inherit the biases of these base libraries (e.g., cultural biases of the "everyday" objects in PartNet and biases toward small plastic 3D printable objects in Thingi10K).

There are a number of directions for future work that are made possible by our dataset. In computer graphics, using our dataset facilitates the development of real-time example-based fracture algorithms [50]. In archaeology [9], our dataset can be used to study the problems of missing fractured pieces [63] or distorted parts. Another interesting direction would be studying geometric shape assembly in few-shot [57] and zero-shot [56] settings, since data is often scarce or even not available in real world. In robotics, our dataset facilitates the development of sequential decision-making algorithms [12] with task-oriented grasping [64] to achieve robotic assembly. We hope that releasing our dataset and a testbed that includes all three baseline methods will allow multiple communities to form discussions and development around this practical yet challenging fracture reassembly problem.

## Acknowledgments

This project is funded in part by NSERC Discovery (RGPIN2017–05235, RGPAS–2017–507938), New Frontiers of Research Fund (NFRFE–201), the Ontario Early Research Award program, the Canada Research Chairs Program, a Sloan Research Fellowship, the DSI Catalyst Grant program and gifts by Adobe Inc. Silvia Sellán is funded in part by an NSERC Vanier Scholarship and an Adobe Research Fellowship. Animesh Garg is supported by CIFAR AI chair, NSERC Discovery Award, University of Toronto XSeed Grant and NSERC Exploration grant. We would like to acknowledge Vector institute for computation support. We thank Xuan Dam, John Hancock and all the University of Toronto Department of Computer Science research, administrative and maintenance staff that literally kept our lab running during very hard years.

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
