# OpenReview forum: "Breaking Bad: A Dataset for Geometric Fracture and Reassembly"
_NeurIPS.cc/2022/Track/Datasets_and_Benchmarks — NeurIPS 2022 Datasets and Benchmarks _

### Official Review · Reviewer_GGsq · 2022-07-25
**A nice dataset, but several improvements could be make, especially for the baseline design**

**Rating:** 5
**Confidence:** 4
**Correctness:** The claims are all correct to the bes…

**Strengths:**

1. It is the first large-scale dataset for reassembly based on physical simulation, which could be helpful for domains like manufacturing, archaeology, and computer vision.

2. In the meanwhile, the efficiency of generating desctructed compenets is high, which provides the possibility to further increase the amount of the data. Also, the authors proposed a lossless compression for the data access, which could help the use and spread of the dataset in practice.



**Weaknesses:**

1. My major concern about this paper is the baseline adaptation, considering the poor performance when the number of fractures is more than two. The authors should have more analysis for the baseline methods.
Other questions:
First, can the author consider using other 3D data types, like signed distance function (SDF) for the learning purpose? As it may have a better ability than just using point cloud to learn complex shapes. Second, though using the Euler angle to parameterize the rotation is intuitive, it might be problematic when the rotation angle is large. Using 6DOF representation[1] or Bingham rotation[2] might help improve the performance. And RMSE and MAE for the Euler angle might also be problematic since the Euler anger is discontinuous.

2. The authors should mention more about the physical simulation assumption to help the user understand under what assumptions and conditions are these data generated. I suggest the authors condense Line 115-128 and refer them to the original paper for replacement.

3. The author should prove the simulation is a good approximation to the real-world destruction process, so it is meaningful for the users to use it.

4. It would be better if the authors can consider different materials for the objects.

5. For the simulation, should the balancing weight omega be varied during the data generation process? Why did the author fix 0.001 here?

Last, I am not sure if the contribution of this paper is substantial, since, for the data generation, the authors are using an existing algorithm.

[1] Zhou, Yi, et al. "On the continuity of rotation representations in neural networks." Proceedings of the IEEE/CVF Conference on Computer Vision and Pattern Recognition. 2019.
[2] Peretroukhin, Valentin, et al. "A Smooth Representation of SO(3) for Deep Rotation Learning with Uncertainty." Proceedings of Robotics: Science and Systems. 2020

**Additional Feedback:**

Can the authors consider a loss that can constrain the connection of each fragment? From Fig. 5 we can see many parts are not connected.



**Clarity:**

The paper is overall well written, and the logic of this paper is also sound to me.
I have one question regarding the writing:
First, for Tab.2, if 25/50/75-th is representing the data sample at the 25/50/75 percentile, why do the authors divide it by the number of objects?


**Documentation:**

I think all the data, documents, and licenses are accessible from the paper.

**Ethics:**

I cannot think of any ethical problem for such a dataset.

**Relation To Prior Work:**

The authors are using an existing simulation algorithm to generate a dataset. The related work part is clear.

**Summary And Contributions:**

This paper proposed a dataset for geometric fracture and reassembly, based on 3D geometry. Different from some dataset that considers the assembly as a graph relationship for different semantic parts, this dataset provides the fragments that are generated from a destruction process for an object.

For the generation of these fragments, the authors use an existing algorithm to simulate the destruction process. Also, the authors benchmark three deep learning baseline methods, showing further effort is needed in this direction.

I think the motivation of this dataset is strong, as it could help many potential domains like manufacturing, archaeology, and computer vision.  However, some questions should be answered and some issues should be addressed before the paper can be accepted.

---

> ### Author Response · Authors · 2022-08-19
> **Responses (1/2)**
>
> **Q1:**
> > My major concern about this paper is the baseline adaptation, considering the poor performance when the number of fractures is more than two. The authors should have more analysis for the baseline methods.
>
> **A:**
> In semantic part assembly, all part shapes have clear semantic meanings and semantic part assembly can be achieved by leveraging such priors. To achieve this, existing shape assembly methods use PointNet to learn a global shape feature for each part. In contrast, geometric shape assembly has to rely on learning local features for local surface matching. Existing semantic part assembly baselines do not learn local features and therefore are not able to tackle the geometric shape assembly task. We have included these remarks in L273-276 in the revision.
>
> ---
>
> **Q2:**
> > Can the author consider using other 3D data types, like signed distance function (SDF) for the learning purpose? As it may have a better ability than just using point cloud to learn complex shapes.
>
> **A:**
> Indeed, we agree that SDFs can be a better representation for modeling shapes compared to point clouds. However, we are not aware of any existing methods that use SDFs for shape assembly. As a submission to the dataset and benchmark track, we only benchmark existing point-cloud-based methods on our dataset.
>
> ---
>
> **Q3:**
> > Though using the Euler angle to parameterize the rotation is intuitive, it might be problematic when the rotation angle is large.  And RMSE and MAE for the Euler angle might also be problematic since the Euler anger is discontinuous.
>
> **A:**
> Euler angle as rotation representation: In our implementation, we did use quaternion to represent rotation. We follow [7] and use Euler angle to quantify performance. In Section 3.2 of [a], the MSE of Euler angle is a valid metric in SO(3) under some constraints. We followed [a] and implemented such constraints in our evaluation code. Therefore, RMSE and MAE are valid metrics.
>
> [a] Huynh, Du Q. "Metrics for 3D rotations: Comparison and analysis." Journal of Mathematical Imaging and Vision 35.2 (2009): 155-164.
>
> ---
>
> **Q4:**
> > Using 6DOF representation or Bingham rotation might help improve the performance.
>
> **A:**
> We also tried the 6-DoF representation but did not observe performance improvement.
>
> ---
>
> **Q5:**
> > The authors should mention more about the physical simulation assumption to help the user understand under what assumptions and conditions are these data generated. I suggest the authors condense Line 115-128 and refer them to the original paper for replacement.
>
> **A:**
> We respectfully disagree to condense L115-128. These lines describe the assumption and conditions of how the fractures are generated. We have added a sentence in L136-137 that refers the reader to [51] for more details about the fracture simulation algorithm.
>
> ---
>
> **Q6:**
> > The author should prove the simulation is a good approximation to the real-world destruction process, so it is meaningful for the users to use it.
>
> **A:**
> We chose the fracture simulation algorithm because it uses a physical model to simulate the most likely fracture patterns for a given object. This algorithm is not a contribution of our work and we believe that its superiority when compared to other similarly scalable fracture algorithms is proven in Sellan et al. [51].

---

> > ### Author Response · Authors · 2022-08-19
> > **Responses (2/2)**
> >
> > **Q7:**
> > > It would be better if the authors can consider different materials for the objects.
> >
> > **A:**
> > We agree with the reviewer that material diversity is key to generalizability. Even if there is no known empirical connection to real-world physical material properties yet, the simulation parameter $\sigma$ relates to the material cohesion of the object. We randomize this value during our dataset generation process to thus simulate different brittle materials. We have clarified this in the revised Sections 3 and 4.1.
> >
> > We are excited about other future work incorporating different materials into our generation pipeline without sacrificing scale. For example, by using a neural network to predict the material of each given object before processing, or semantically segmenting it into different materials. We have added references to these possibilities in Section 7.
> >
> > ---
> >
> > **Q8:**
> > > For the simulation, should the balancing weight $\omega$ be varied during the data generation process? Why did the author fix 0.001 here?
> >
> > **A:**
> > As pointed out in Sellán et al. [51], the $\omega$ weight has no effect on the solution as long as it is low enough to not produce elastic modes (see Figure 12 in [51]). The model by Sellán et al. [51] is appropriate for brittle fracture only when $\omega$ is small enough, so we fix it at 0.001. We have added a sentence to Section 3 to clarify this.
> >
> > ---
> >
> > **Q9:**
> > > I am not sure if the contribution of this paper is substantial, since, for the data generation, the authors are using an existing algorithm.
> >
> > **A:**
> > As a submission to the datasets and benchmarks track, our contributions are a dataset of fractured objects at a massive scale and benchmarks for state-of-the-art shape assembly algorithms. The reviewer is correct that we use an existing algorithm for the dataset generation, which we consider to be outside of our contribution.
> >
> > ---
> >
> > **Q10:**
> > > I have one question regarding the writing: First, for Tab.2, if 25/50/75-th is representing the data sample at the 25/50/75 percentile, why do the authors divide it by the number of objects?
> >
> > **A:**
> > The #FP/#O notation stands for the number of fractured pieces per object as described in L177-178. Therefore, this statistic shows the distribution of how many fractures an object is broken into in our dataset.
> >
> > ---
> >
> > **Q11:**
> > > Can the authors consider a loss that can constrain the connection of each fragment? From Fig. 5 we can see many parts are not connected.
> >
> > **A:**
> > The baseline methods are not designed to perform local matching between pieces, but to predict a pose for each fragment. Therefore, the connections between pieces are not guaranteed. To constrain the connection between fragments, the model has to first identify the connecting interfaces between pieces and then predict how the interfaces are connected together. As a submission to the dataset and benchmark track, we only consider existing methods. We believe that such a novel network architecture design is beyond the scope of our paper. We hope that future work can address this issue by incorporating the reviewer’s suggestions.

---

### Official Review · Reviewer_yT69 · 2022-07-25
**The introduced Breaking Bad dataset is interesting and valuable for 3D community.**

**Rating:** 7
**Confidence:** 4
**Clarity:** Yes, the paper is well written.

**Strengths:**

* The proposed geometric fracture dataset is large-scale, physical realistic and considers diverse breaking patterns in different level of scales. Breaking Bad consists of 10,474 shapes, with 100 breakdown patterns in each shape. The decomposition is geometric-based, which is different from the semantic-based datasets, e.g. PartNet. The produced fractures are closer to the realistic scenario.
* The generation of the dataset is fast. Besides, the dataset is compressed for efficient storage, which can be easily accessed.
* The benchmark for 3D geometric shape assembly is interesting. The results show that current SOTA semantic assembly methods fail to assemble well geometrically. There is still a large space for improvement. Besides, the benchmark analyzes the influence brought by the number of fractured pieces. It shows that a larger number of pieces leads to a more challenging  situation. The dataset is worthwhile to further explore.

**Weaknesses:**

* The case study only includes 3D geometric shape assembly. I hope to see some possible benchmarks for downstream tasks to validate its value, such as tasks like missing fractured pieces study mentioned in the future work.
* The validation of the dataset should be included. I believe that there exists some fractures generated are not truly realistic. Please explain how to deal with these pieces and provide failure case analysis.

**Additional Feedback:**

The Breaking Bad dataset is a valuable dataset for 3D shape assembly and can facilitate the 3D shape geometric analysis. I will consider improving the rating after receiving the responses for the weaknesses.

**Correctness:**

The claims are correct. The dataset is constructed well. The benchmark for fracture assembly is appropriate and performed correctly.

**Documentation:**

The details and documents are provided on the website.

**Ethics:**

There is no ethical concern.

**Relation To Prior Work:**

Yes, the relation to prior work is discussed in the related work.

**Summary And Contributions:**

This paper introduces a large-scale dataset of fractured objects in a physically realistic scenario. The dataset consists of 20 daily object categories and artifacts with a series of fractures at various scale levels. The simulation of fracture modes is fast and produces impact-dependent fracture patterns. Three SOTA deep learning based shape assembly methods for semantic assembly, are benchmarked on the proposed dataset. These methods are hard to assemble fractures produced by impacts. The dataset facilitates several research works and is useful in different scenarios.

---

> ### Author Response · Authors · 2022-08-19
> **Responses**
>
> **Q1:**
> > The case study only includes 3D geometric shape assembly. I hope to see some possible benchmarks for downstream tasks to validate its value, such as tasks like missing fractured pieces study mentioned in the future work.
>
> **A:**
> In the paper, we benchmarked our dataset on the geometric shape assembly task, since this is one of the applications our dataset can be used for and there are existing shape assembly methods available. While our dataset could be used for studying the missing fractured pieces setting, we believe that benchmarking in this setting is beyond the scope of this paper and leave it as future work. We hope that our dataset, which is the main contribution of our work, opens the door for researchers to experiment with different tasks and settings in the future.
>
> ---
>
> **Q2:**
> > The validation of the dataset should be included. I believe that there exist some fractures generated that are not truly realistic. Please explain how to deal with these pieces and provide a failure case analysis.
>
> **A:**
> We selected Sellán et al.’s algorithm [51] optimizing for scalability and physical realism, as it uses a physical model to compute a shape’s most natural breaking patterns. Like any physical model, it has its limitations (see Section 7). In our revised L167 onwards, we explain how we dealt with a specific failure case of the simulation algorithm (it sometimes produces too many fracture pieces).

---

### Official Review · Reviewer_xdoY · 2022-07-27
**A simulated dataset for benchmarking geometric fracture and reassembly**

**Rating:** 9
**Confidence:** 4
**Clarity:** Yes. The paper is well written and ea…

**Strengths:**

1. This paper successfully applies a fracture simulation algorithm to creating a large-scale fractured object dataset. The dataset is potentially helpful for studying the shape assembly task.
2. The data collection and processing pipeline is well explained, making it possible for other researchers to expand the dataset when necessary.
3. Good dataset analyses and visualizations are provided.
4. A feasible data compression practice is used so it is much easier to download the dataset.
5. The authors develop a benchmark for the shape assembly task with appropriate data splits and evaluation protocols, which can significantly facilitate the research on this problem.

**Weaknesses:**

1. As the authors point out in the paper, this dataset is strongly biased by the underlying fracture simulation method. Although the authors deliberately select object categories that fit the simulator’s characteristics, there is still some domain gap between the simulation and the reality. Given a shape assembly method that works well on the proposed dataset, it is not clear how it can generalize to real-world data. It would be nice if the authors can discuss this domain gap and provide insights on what use cases this dataset is not suitable for.
2. The authors may need to justify the usage of the specific simulation method. Computational efficiency does not seem to be a valid advantage for dataset generation, as everything can be processed offline. If there is a computational expensive method that can simulate more realistic fracture, it should be the preferable one. Moreover, it might be even better if multiple simulation methods are used for data generation, as it may be able to mitigate the data bias.

**Additional Feedback:**

See "Weaknesses".

**Correctness:**

Yes. The claims made in the submission are correct. The dataset is constructed in a sound way. The evaluation methods and experiment design are appropriate and performed correctly.

**Documentation:**

Yes. There is sufficient detail on data collection and organization, availability and maintenance, and ethical and responsible use. There is also sufficient detail to support reproducibility.

**Ethics:**

No. There is no ethical concern.

**Relation To Prior Work:**

Yes. The difference of this work from previous works is clearly discussed.

**Summary And Contributions:**

This paper proposes a large-scale fractured object dataset by applying an off-the-shell fracture simulation algorithm on existing 3D object datasets. The objects are assumed to be made from solid isotropic materials. For each 3D object in the dataset, 20 fracture modes are pre-computed and 80 fracture patterns are simulated. The authors show the value of the proposed dataset by benchmarking learning-based shape assembly methods.

---

> ### Author Response · Authors · 2022-08-19
> **Responses**
>
> **Q1:**
> > As the authors point out in the paper, this dataset is strongly biased by the underlying fracture simulation method. Although the authors deliberately select object categories that fit the simulator’s characteristics, there is still some domain gap between the simulation and reality. Given a shape assembly method that works well on the proposed dataset, it is not clear how it can generalize to real-world data. It would be nice if the authors can discuss this domain gap and provide insights on what use cases this dataset is not suitable for.
>
> **A:**
> We selected Sellán et al.'s algorithm [51] optimizing for scalability and physical realism. As opposed to procedural methods with similar runtimes, the algorithm we use is derived from a physical strain model. As we detail in L308 onwards, this choice limits the shapes and materials our dataset is suitable for. Notably, we limit ourselves to brittle materials where fractures can be approximated as happening instantaneously and the object does not noticeably deform prior to breaking.
>
> ---
>
> **Q2:**
> > Deliberately select object categories that fit the simulator’s characteristics.
>
> **A:**
> We respectfully disagree. We selected object categories from PartNet and Thingi10K. While the objects from PartNet are selected from some particular categories, we did not do so for objects in Thingi10K. This selection choice is orthogonal to the choice of the simulator.
>
> ---
>
> **Q3:**
> > The authors may need to justify the usage of the specific simulation method. Computational efficiency does not seem to be a valid advantage for dataset generation, as everything can be processed offline. If there is a computationally expensive method that can simulate a more realistic fracture, it should be the preferable one. Moreover, it might be even better if multiple simulation methods are used for data generation, as it may be able to mitigate the data bias.
>
> **A:**
> Realistic state-of-the-art physical simulation of the fracture process requires extremely fine spatial and temporal discretization, with runtimes many orders of magnitude higher than our chosen algorithm (see, e.g., Table 2 in “Adaptive Tetrahedral Meshes for Brittle Fracture Simulation” by Koschier et al. 2015). This makes even offline dataset generation impracticable at our novel desired scale (generating one million fractures, e.g., with the method by Koschier et al. and their own average reported timings would take between 200 and 400 days).
>
> Among procedural algorithms whose runtimes would make the dataset generation task feasible, we choose the fracture simulation algorithm in [51] because it computes a set of orthogonal (i.e., sufficiently different) most natural (i.e., likely) fracture patterns. We have made these reasons more explicit in the last paragraph of Section 2. We agree with the reviewer that including fractures using other and future not-yet-discovered methods in our dataset would mitigate any bias introduced by the algorithm. We have added this as a potential future work in Section 7.

---

### Official Review · Reviewer_kmLT · 2022-07-27

**Rating:** 8
**Confidence:** 3
**Correctness:** Yes.
**Clarity:** Yes.

**Strengths:**

- I think this is an interesting and real problem to trackle. While prior works only focus on assemblying 3d shapes with semantically well-defined parts, it brings a new challenge.
- This is a large-scale datasets with more than 1M 3D models. These 3D models are open-source.
- The paper is well-written with fantastic figures.


**Weaknesses:**

I don't see any obvious weakness. I agree the chosen fracture simulation method brings fundamental limitations. And the dataset is synthetic instead of real. However, as the first attempt, I still think it's valuable to our community.

**Additional Feedback:**

N/A

**Documentation:**

Yes. The website provides good 3D visualization of a subset of breaking bad.

**Ethics:**

No. I only worry about the license of 3D models but it looks like authors have addressed it well.

> We gather our base models following the licenses specified in each of the source datasets: the MIT
license in the PART N ET dataset and a variety of open-source licenses in the T HINGI 10 K dataset (see
Figure 12 in [65]). We release each model in our dataset with an as-permissive-as-possible license
compatible with its underlying base model and all code under the MIT license.

**Relation To Prior Work:**

Yes.

**Summary And Contributions:**

The paper proposes Breaking Bad, a new dataset containing over one million fractured objects. While most object part datasets use semantics to disassemble objects, a physical fracture does not follow semantics. The paper samples 3D models from Thingi10K and PartNet, simulate fracture patterns and build the whole new dataset. It brings new challenges and benchmarks for 3D geometric shape assembly.

---

> ### Author Response · Authors · 2022-08-19
> **Response**
>
> **Q:**
> > I don't see any obvious weakness. I agree the chosen fracture simulation method brings fundamental limitations. And the dataset is synthetic instead of real. However, as the first attempt, I still think it's valuable to our community.
>
> **A:**
> We are thankful to the reviewer for recognizing our work’s value to the community. We selected Sellán et al.’s algorithm [51] optimizing for scalability and physical realism. As we detail in Section 7 of the original submission, this means inheriting some of its limitations. We hope that our dataset will inspire interesting future works.

---

### Official Review · Reviewer_HUdK · 2022-07-28
**Challenging Dataset for Shape Assembly and Geometric Understanding Task**

**Rating:** 7
**Confidence:** 2

**Strengths:**

The dataset is a good contribution towards benchmarking methods for shape assembly and towards geometric shape understanding. As demonstrated in the paper, the complexity of the shape assembly task especially when the fracture results in multiple pieces, has been demonstrated via benchmarking of the selected baseline architectures.

The dataset defines a challenging task where the inductive bias and applicable priors are weaker since the fractures are geometric rather than semantic. This might help push current state of the learning algorithms.



**Weaknesses:**

Not necessarily a weakness, but the choice of objects in the dataset which includes everyday objects seem to imply open-ended applicability/use-case rather than a focused narrow domain of subtask, for example, say a femur fracture dataset.

**Additional Feedback:**

The design of the dataset seems to be open-ended in its applicability/use-case rather than a focused narrow domain of subtask. Maybe the method used to develop this dataset can potentially be useful in the medical imaging community where data acquisition can be expensive and synthetic datasets may help (for example, for generating femur fracture dataset).

**Clarity:**

Yes, the paper is well written and is organized such that the text flows naturally.

**Correctness:**

Yes. Although the dataset uses meshes from already available datasets, the said contribution is useful and correct.

**Documentation:**

The authors have provided hosting plan, long term data accessibility and datasheet for datasets. The code is also available.

**Relation To Prior Work:**

Yes, relevant literature is discussed.

**Summary And Contributions:**

The paper contributes a large dataset of simulated fractures on various mesh objects assembled from publicly available datasets. Their earlier contribution of generating more accurate fractures than procedural methods without (very computation intensive) full physical simulation was used to generate plausible fractures. Various possible use-cases of the dataset has been indicated.

Benchmarking of various architectures for shape assembly as well as the difficulty of the task has been demonstrated.

Compared to earlier shape assembly datasets, their contribution is in generating realistic fractures which means the fractured pieces themselves may not represent semantic part individually. This presents a more challenging task whereby understanding of semantic object composition only may not be sufficient.

---

> ### Author Response · Authors · 2022-08-19
> **Response**
>
> **Q:**
> > Not necessarily a weakness, but the choice of objects in the dataset which includes everyday objects seem to imply open-ended applicability/use-case rather than a focused narrow domain of subtask, for example, say a femur fracture dataset.
>
> **A:** We thank the reviewer for recognizing our dataset’s open-ended applicability, indeed, due to the wide variety of object categories. Moreover, our data generation process is procedural and can be applied to objects in other problem domains (e.g., femurs). Our data generation code and instructions are publicly available on the project page. We hope that specialized communities (e.g., healthcare and computational archaeology) can benefit from using our code to create customized datasets.

---

> > ### Comment · Reviewer_HUdK · 2022-08-27
> > **Satisfied with the update**
> >
> > I would like to thank the authors for the response and updating the paper. I am satisfied with the update and addresses most of my inquiries.

---

### Author Response · Authors · 2022-08-19
**General response**

We thank the reviewers for providing insightful feedback. We have addressed each of the questions in detail and revised the manuscript. For ease of review, we highlight the revised text in blue in the revised manuscript.

We would like to restate our main contributions: a large-scale dataset of fractured objects and benchmarks for existing shape assembly algorithms. To produce over one million synthetic fractures at a reasonable cost, we were restricted to the class of pre-fracture algorithms, which compute a fracture pattern for a given object once and then efficiently simulate the fracture pieces resulting from many different impacts. Among these, we selected Sellán et al. [51] for their physical realism and their capacity for producing maximally different (orthogonal) fracture patterns.

Furthermore, we would like to note that as a submission to the dataset and benchmark track, we only consider methods that were published before the paper submission deadline and their official implementations were publicly available at the time, i.e., the three methods considered in Table 3. Notably, we are not claiming novelty in methods for fracture or solutions to the shape assembly problem in this paper. In addition, while our dataset can potentially be used for other applications, we benchmark existing methods on the geometric shape assembly task and treat other interesting applications (those listed in Section 7) as future work.

---

### Author Response · Authors · 2022-08-24
**Author/reviewer disucssions**

Dear Reviewers,

Thank you for providing insightful feedback to help us improve our work. We have addressed all the questions in detail and revised the manuscript. Please have a look at our response to each question and the revised manuscript. Please let us know if there is anything that needs further clarifications. Thank you.

Authors

---

### Meta-Review · Program_Chairs · 2022-09-16

**Recommendation:** Accept
**Confidence:** 4

**Metareview:**

The paper contributes towards generating realistic fractures where the fractured pieces themselves may not represent semantic part individually. This presents a challenging task which requires understanding beyond semantic object composition. The reviews are aligned in the largely positive reviews (4 positive, 1 negative) for the paper and the authors have successfully addressed the weaknesses raised by the reviewers.
I believe that this is an important contribution towards the study of geometric fracture and reassmebly and should be accepted.

---

### Decision · Program_Chairs · 2022-09-16

Accept